# pH-Responsive Drug Delivery and Imaging Study of Hybrid Mesoporous Silica Nanoparticles

**DOI:** 10.3390/molecules27196519

**Published:** 2022-10-02

**Authors:** Zhongtao Li, Jing Guo, Guiqiang Qi, Meng Zhang, Liguo Hao

**Affiliations:** 1Department of Molecular Imaging, School of Medical Technology, Qiqihar Medical University, Qiqihar 161006, China; 2Department of Molecular Imaging, The First Affiliated Hospital of Qiqihar Medical University, Qiqihar 161041, China; 3Laboratory Animal Center, Qiqihar Medical University, Qiqihar 161006, China

**Keywords:** mesoporous silica, pH-responsive, magnetic resonance imaging, drug delivery, controlled release

## Abstract

A system of pH-responsive and imaging nanocarriers was developed using mesoporous silica nanoparticles (MSNs), in which gadolinium (Gd) was doped through in situ doping (Gd_2_O_3_@MSN). Sodium alginate (SA) was attached to the surfaces of the amino groups of MSNs (NH_2_-Gd_2_O_3_@MSN) through the electrostatic adsorption between the amino groups and the carboxyl groups with the formation of hybrid SA-Gd_2_O_3_@MSN nanoparticles (NPs). The SA-coated NPs were spherical or near-spherical in shape with an average size of nearly 83.2 ± 8.7 nm. The in vitro drug release experiments of a model rhodamine B (RhB) cargo were performed at different pH values. The result confirmed the pH-responsiveness of the nanocarriers. The results of the cytotoxicity studies indicated that the SA-Gd_2_O_3_@MSN NPs were not cytotoxic by themselves. The results of the in vivo safety evaluation and the hemolysis assay confirmed that the system is highly biocompatible. It is noteworthy that the T1 contrast of the system was significantly enhanced by the Gd, as indicated by the result of the MR imaging. This study confirms that the synthesized hybrid nanosystem is promising for pH-responsive drug delivery and MR imaging for cancer diagnosis and treatment.

## 1. Introduction

Nanotechnology has supplied a novel arsenal to drug delivery systems over the past few years, which has inspired a novel field, i.e., nano-drug delivery systems (NDDSs). The benefits of using an NDDS versus conventional chemotherapy involve the increased bioavailability, solubility of the drugs, and pharmacokinetics behavior in vivo, thereby reducing the side effects and increasing the curative effect of the therapies [1]. The common NDDSs include polymers, dendrimers, inorganic nanoparticles and organic–inorganic hybrid nanosystems [2,3,4,5,6]. Among the above nanosystems, mesoporous silica nanoparticles (MSNs) have attracted the attention of numerous researchers and have become the highly popular drug nanocarriers for their facile production, tunable uniform pore size, large pore volume, high surface area, and great drug loading capacity [7,8]. A dedicated mesoporous silica NDDS design has been extensively used [9,10] since silica-based materials have been considered safe by the United States Food and Drug Administration.

Additionally, research on innovative multifunctional nanosystems with the combination of imaging and therapy has become a hotspot. Magnetic resonance imaging (MRI), nuclear medicine imaging (NMI), and ultrasound (US) techniques have been extensively used [11,12,13,14] in the context of medical imaging modalities (e.g., computed tomography (CT)). To be specific, MRI is a popular imaging technology due to its high-resolution morphological capabilities, non-invasiveness, and lack of risk of radiation damage [15,16]. More excitingly, incorporating the MRI contrast agents (e.g., gadolinium oxide (Gd_2_O_3_)) into nanosystems can extend the blood circulation time while endowing the nanosystems with MRI visibility by taking the shortened longitudinal proton relaxation time of the surrounding hydrogen protons [17,18].

To achieve the release of the drug in the targeted diseased tissues, several surface-functionalized coatings as gatekeepers are modified onto the MSN carriers to block the drug in the pore channels [19,20,21,22]. Various types of stimuli could be used to open the gatekeepers, such as pH [23,24], enzymes [25], redox [26,27], light [28], ultrasound [29], and temperature [30]. Among them, pH-responsive systems have gained wide attention because of the distinct pH values in different tissues (e.g., healthy or tumor tissues) [31]. For this reason, a pH-dependent NDDS has been designed for pH-triggered drug release.

Sodium alginate (SA), an anion- and water-soluble linear polysaccharide, comprises 1-4-linked β-D-mannuronate (M) and α-L-guluronate (G) [32]. SA has been widely employed in pharmaceutical excipients for its fascinating advantages (including its safety, biocompatibility, and biodegradability [33,34,35,36]). SA is relatively low-cost and easily available via the isolation from cell walls of brown algae (*Ascophyllum nodosum* and *Laminaria digitate*) as compared with polymers and polyelectrolytes [33,37]. Moreover, the existing research has reported that SA is capable of making nanoparticles more dispersible and stable [38]. Accordingly, the polysaccharide SA was selected as a capping agent for its high biodegradability and appropriate size to cover the MSNs so as to control the drug release.

A pH-responsive delivery and imaging nanosystem was developed in this study (Figure 1). In the developed system, the MSNs (Gd_2_O_3_@MSN) were prepared by doping Gd^3+^. The polysaccharide SA was attached onto the surfaces of the MSNs through electrostatic adsorption, which served as the pH-sensitive end-cap, while providing better biocompatibility. RhB was adopted as the model drug to study the loading capacity of the MSNs. Moreover, in vitro drug release experiments were performed to examine the pH-responsive release under acidic conditions [39,40]. Furthermore, the Gd^3+^-incorporated MSNs can serve as MRI contrast agents to monitor drug delivery.

## 2. Results and Discussion

### 2.1. Synthesis and Characterization

Nanotheranostic platforms are capable of efficiently integrating the processes of diagnosis and therapeutic moiety. A hybrid nanocarrier was successfully synthesized for MR imaging and drug loading in this study. Figure 1a illustrates the synthesis of RhB-loaded SA-Gd_2_O_3_@MSN hybrid systems. First, Gd^3+^-doped MSN nanoparticles (Gd_2_O_3_@MSN) were prepared. Subsequently, the surfaces of Gd_2_O_3_@MSN were grafted with APTES to become amino-functionalized Gd_2_O_3_@MSN (NH_2_-Gd_2_O_3_@MSN). Next, SA was modified onto the NH_2_-Gd_2_O_3_@MSN surfaces through the electrostatic interaction between the amino group of NH_2_-Gd_2_O_3_@MSN and the carboxyl group in SA [38,41,42]. As depicted in the high-resolution TEM image in Figure 1a, the average diameter of the Gd_2_O_3_@MSN reached 77.5 ± 9.4 nm, and the Gd_2_O_3_@MSN exhibited a clearly defined pore structure and regular morphology. Compared with the smooth surface of Gd_2_O_3_@MSN, the grafted polysaccharide SA-Gd_2_O_3_@MSN surface (Figure 1b) was blurred, indicating that the SA covered the MSN carrier. EDS mapping images (Figure 1d) showed the coexistence of Gd, Si, and O in the Gd_2_O_3_@MSN NPs. Afterward, the elemental content was further analyzed using the EDS spectrum (Figure 1c), and the exact content of gadolinium was identified as 5.3 wt% through an ICP-MS (Agilent 720 ES, Shanghai, China) analysis. The above results significantly indicated that the Gd element was successfully loaded into the MSNs. 

The successful synthesis of all products was achieved using a variety of methods. The zeta potential of NH_2_-Gd_2_O_3_@MSN underwent a great change after functionalization, varying from −21.9 to + 28.7 mV (Figure 2a), which indicated the addition of the amine groups to the surface of Gd_2_O_3_@MSN. Upon the encapsulation of SA, the surface charge of the SA-Gd_2_O_3_@MSN was significantly decreased to −21.1 mV, which could be explained by the considerable carboxyl groups in the SA polysaccharide [43]. The hydrodynamic diameter and PDI values of Gd_2_O_3_@MSN, NH_2_-Gd_2_O_3_@MSN, and SA-Gd_2_O_3_@MSN were obtained in ddH_2_O (Figure 2b,c). The diameters of Gd_2_O_3_@MSN and NH_2_-Gd_2_O_3_@MSN were 168.7 nm and 193.6 nm, respectively. The hydrodynamic diameter increased to 233.4 nm after the addition of SA on the surfaces of NH_2_-Gd_2_O_3_@MSN to form SA-Gd_2_O_3_@MSN. The PDI of SA-Gd_2_O_3_@MSN was smaller than that of NH_2_-Gd_2_O_3_@MSN. In addition, the diameters obtained via TEM were smaller than those examined via DLS, probably due to the slight aggregation and hydration layer of NPs under aqueous conditions [44,45]. Figure 2d presents the FT-IR spectra of Gd_2_O_3_@MSN, NH_2_-Gd_2_O_3_@MSN, SA-Gd_2_O_3_@MSN, and SA. The produced peaks were at 1080 cm^−1^, corresponding to Si-O-Si vibration. Next, the presence of amine groups (NH_2_-) on the NH_2_-Gd_2_O_3_@MSN surfaces was indicated by the appearance of the characteristic peak (N-H) at 2900 cm^−1^ [46]. After SA was added to the NH_2_-Gd_2_O_3_@MSN surface, there was an increase in the adsorption peaks at 1400 cm^−1^, due to the vibration of -COOH in the SA [29]. All of the results confirm the successful preparation of SA-Gd_2_O_3_@MSN.

In addition, the pore size distributions and surface areas of a range of modified MSN nanoparticles were examined through the N_2_ adsorption–desorption measurements at 77 K (Figure 3 and Table 1). In brief, the curve of NPs was analyzed using the Brunauer–Emmett–Teller method. The surface area and pore volume of Gd_2_O_3_@MSN were 956.18 m^2^/g and 0.74 cm^3^/g, respectively. For SA-Gd_2_O_3_@MSN, the above parameters were reduced to 672.09 m^2^/g and 0.45 cm^3^/g, respectively, indicating that SA coated the MSN surfaces.

### 2.2. Release Experiments from SA-Coated Mesoporous Silica

A series of release experiments were performed to examine the pH-responsiveness of the hybrid mesoporous silica, using a model fluorescent RhB as the cargo. The successful RhB loading on the nanocarrier was evidenced by the FT-IR spectra of the samples (Appendix A). The LC of SA-Gd_2_O_3_@MSN-RhB was 18.9 μg/mg. To simulate the physiological conditions, the RhB-loaded and SA-coated NPs were incubated at 37 °C, using three pH values (7.4, 5.5, and 4.5), corresponding to the normal blood, endosome, and lysosomal environments, respectively (Figure 4a) [47,48]. At neutral pH (7.4), 18.2% of the RhB was released after 48 h of incubation. However, the RhB release contents were increased to 52.1% at pH 5.5 and to 81.2% at pH 4.5, respectively. These results were attributed to the weakened electrostatic interaction between the negatively charged carboxylic acid group of the SA and the positively charged MSNs, leading to the opening of the mesoporous MSNs [49,50,51,52]. The above result suggests that the release of RhB is significantly pH-dependent.

In addition, the 48 h cumulative drug release rates from MSN NPs without SA coating at pH 7.4 and pH 5.5 were higher than the cumulative drug release rate from SA-Gd_2_O_3_@MSN-RhB NPs at 48 h, which clearly showed that SA is capable of hampering the dye release (Appendix A). 

### 2.3. Assessment of the T1 Relaxivity

The results of this study confirmed the pH-triggered release of RhB from hybrid mesoporous silica. Next, the potential use of SA-Gd_2_O_3_@MSN as an MRI contrast agent was examined using the 0.5 T NMI20 Analyst NMR system. The relaxivity (r1) rates of Gd_2_O_3_@MSN and SA-Gd_2_O_3_@MSN were respectively obtained as 52.48 mM^−1^s^−1^ (R^2^ = 0.9916) and 11.91 mM^−1^s^−1^ (R^2^ = 0.9911) at 0.5 T, which were significantly higher than for the clinically applied Gd-DTPA (r1 = 4.52 mM^−1^s^−1^, R^2^ = 0.9961), suggesting that the Gd_2_O_3_@MSN may serve as an MRI contrast agent (Figure 4b). The lower relaxation of SA-Gd_2_O_3_@MSN compared with Gd_2_O_3_@MSN was tentatively attributed to the SA coating preventing the water molecules from entering the mesoporous structure, resulting in less water molecules being bound to the Gd center [53,54].

### 2.4. Colloidal Stability

To compare their colloidal stability levels, Gd_2_O_3_@MSN, NH_2_-Gd_2_O_3_@MSN, and SA-Gd_2_O_3_@MSN were suspended in ddH_2_O, saline, and RPMI 1640 medium supplemented with 10% FBS at 37 ℃ for 24 h. As depicted in Appendix A, SA-Gd_2_O_3_@MSN remained stable in ddH_2_O, saline, and the medium for over 24 h without precipitation, while the structural integrity was maintained. In contrast, Gd_2_O_3_@MSN and NH_2_-Gd_2_O_3_@MSN quickly flocculated in the respective buffers. The hydrodynamic diameter of SA-Gd_2_O_3_@MSN was further examined using DLS, and the results indicated that no significant size changes occurred in the respective buffers (Figure 5). Afterward, free Gd ions in the above solutions were detected using ICP-MS, indicating that SA-Gd_2_O_3_@MSN only released negligible amounts of free Gd^3+^ (Appendix A). The above data suggest that the SA coating can increase the stability of hybrid mesoporous silica.

### 2.5. Biocompatibility and Biotoxicity

A safe nanocarrier system is a prerequisite for further applications. Accordingly, the in vitro cytotoxicity was examined using a CCK-8 assay. As depicted in Figure 6a, the viability of the 4T1 cells and MCF-7 cells was over 80% even after treatment with 200 μg/mL SA-Gd_2_O_3_@MSN, similar to other NPs (Appendix A), suggesting that SA-coated NPs exhibit negligible toxicity. A hemolysis assay was then performed to examine the hemocompatibility of SA-Gd_2_O_3_@MSN. As depicted in Figure 6c, the hemolysis rate was less than 5% for all concentrations, meeting the hemolysis criterion for biomaterials [55,56]. Furthermore, the optical images confirmed that the red blood cells remained intact even after treatment with 800 μg/mL SA-Gd_2_O_3_@MSN (Appendix A).

The SA-Gd_2_O_3_@MSN solution (50 mg/kg) was intravenously injected into healthy Balb/c mice, and the above animals were monitored for 7 days to examine the in vivo biotoxicity from the SA-coated NPs. As depicted in Figure 6b, the body weights of mice in the SA-Gd_2_O_3_@MSN group increased slightly, which were similar to those of the saline group. Then, the results of the H&E staining assay indicated that the major organs of the SA-Gd_2_O_3_@MSN treated mice (e.g., heart, liver, spleen, lung, and kidney) did not exhibit any pathological changes (Figure 6d). No deaths of mice were identified during the experimental process. All of the results suggest that SA-Gd_2_O_3_@MSN exhibits high biocompatibility and can have in-depth applications.

### 2.6. Cellular Uptake of Multifunctional Nanocomposites

To examine the intracellular drug release behaviors of RhB, 4T1 cells were incubated with RhB-loaded and SA-coated NPs for different periods of time and imaged under a CLSM (Figure 7a). As depicted in the CLSM images, the RhB fluorescence (red color) accumulated in the nucleus and cytoplasm of the cells, which was probably due to the pH-induced RhB release from the nanocarriers. Notably, the RhB fluorescence within the cells increased as the incubation time was extended (Appendix A). To further examine the intracellular MRI performance and cellular uptake efficiency, the NPs-treated (100 μg/mL) 4T1 cells were collected and fixed in centrifuge tubes and then examined using a clinical 3.0T MRI scanner (Figure 7b). As depicted in Appendix A, the 4T1 cells after treatment for 24 h exhibited the highest MR signal intensity. The above results suggest that more NPs are successfully taken up by the cells over time.

### 2.7. In Vivo Proof-of-Concept Study: Preliminary In Vivo Short-Term pH Triggering and MR Imaging Performance

To further examine the in vivo pH-triggering of the carrier designed here, the experiments were performed in 4T1 tumor-bearing mice. Fluorescent RhB was selected as the model drug and the release from the carrier was monitored via near-infrared fluorescence imaging (Figure 8a). At 48 h post-injection, a significant fluorescence signal was identified at the tumor site in the SA-Gd_2_O_3_@MSN-RhB group. In addition, the fluorescence signal at the tumor site in the SA-Gd_2_O_3_@MSN-RhB group was stronger than that in the NH_2_-Gd_2_O_3_@MSN-RhB group. In contrast, the fluorescence signal was not identified at the tumor site in the RhB group. During the imaging process, the fluorescent signal in the SA-Gd_2_O_3_@MSN-RhB group was more persistent in the tumor site than NH_2_-Gd_2_O_3_@MSN-RhB and RhB, since the SA-coated NPs can control the release of the drug and the successful pH-triggered dissociation caused by endogenous acids.

For in vivo imaging applications, the T1-weighted MR imaging of the tumor-bearing mice was also performed to verify the feasibility of SA-Gd_2_O_3_@MSN-RhB as an in vivo MRI contrast agent. The NPs was injected into 4T1 tumor-bearing mice. As depicted in Figure 8b,c, the signal intensity at the tumor site was significantly enhanced. The above results suggested that the as-prepared NPs could be used as potential contrast agents for the MR imaging of tumors.

## 3. Materials and Methods

### 3.1. Materials

The cetyltrimethylammonium bromide (CTAB) was provided by Coolaber Science and Technology (Beijing, China). The tetraethylorthosilicate (TEOS) was purchased from Fuchen Chemical Reagents (Tianjin, China). The 3-aminopropyl-triethoxysilane (APTES) and rhodamine B (RhB) were offered by Macklin Biochemical Co., Ltd. (Shanghai, China). The sodium aiginate (SA) and gadolinium(III) chloride hexahydrate (GdCl_3_·6H_2_O) originated from Aladdin Biochemical Technology Co., Ltd. (Shanghai, China). The RPMI 1640 cell culture medium and fetal bovine serum (FBS) were purchased from Gibco (Thermo Fisher Scientific, Waltham, MA, USA). All chemicals used for synthesis were used directly without further purification unless otherwise specified. 

### 3.2. Synthesis of SA-Gd_2_O_3_@MSN

#### 3.2.1. Synthesis NH_2_-Gd_2_O_3_@MSN

The MSNs (Gd_2_O_3_@MSN) were prepared in accordance with our previous research [57]. In brief, a mixture of NaOH (140 mg) and CTAB (500 mg) in ddH_2_O (220 mL) was stirred at 80 ℃ for 1 h. Subsequently, the TEOS (1.5 mL) was added dropwise to the above mixture through appropriate stirring (250 rpm). The GdCl_3_·6H_2_O aqueous solution (20 mL, 5 mg/mL) was rapidly introduced into the mixture after reacting for 2 h at 80 ℃. An additional 1 mL of TEOS was added to the above samples 1 h later. Subsequently, the nanoparticles (NPs) were centrifuged after being stirred for 1 h to collect the samples. Next, the collected samples were dried under vacuum. The resulting NPs were calcined at 600 ℃ for 4 h to remove the CTAB surfactants (Appendix A). Here, 2 mL of APTES and 500 mg of Gd_2_O_3_@MSN were added to 30 mL of ethanol to obtain an amino-functionalized MSN solution (NH_2_-Gd_2_O_3_@MSN), and the solution was then stirred at 30 °C for 36 h [58]. The synthesized NH_2_-Gd_2_O_3_@MSN was collected through centrifugation, washed three times with ethanol, and dried under vacuum for further use.

#### 3.2.2. Synthesis SA-Gd_2_O_3_@MSN

Here, 200 mg of SA was added to 10 mL double-distilled water (ddH_2_O) to prepare the SA solution. Subsequently, 100 mg of NH_2_-Gd_2_O_3_@MSN was dispersed in 30 mL of ddH_2_O. Next, the SA solution (6 mL) was poured into the suspension and then stirred for 24 h at ambient temperature to form the SA-Gd_2_O_3_@MSN nanocomposites through electrostatic absorption. Lastly, the prepared NPs were centrifuged and washed to collect the samples.

### 3.3. Characterization

The morphological features and mesoporous network structure of the as-synthesized NPs were examined using FESEM (field emission scanning electron microscope) images (FEI Talos F200S). The elemental compositions of the samples were determined using elemental mapping and EDS (energy-dispersive spectroscopy) analyses. The BET (Brunauer–Emmett–Teller) surface area, total pore volume, and pore diameter were obtained using nitrogen sorption isotherms (Micromeritics ASAP-2460, Norcross, GA, United States). The FT-IR spectra were examined using a FT-IR spectrometer (FT-IR 6800 JASCO, Marseille, France) in the 400–4000 cm^−1^ region using the KBr pellet technique. The zeta potential and size were recorded with a Nano-z90 Nanosizer (Malvern Instruments Ltd., Worcestershire, UK) at ambient temperature. 

### 3.4. Release Experiments with RhB-Loaded, Polysaccharide-Coated MSN Nanoparticles

The MSNs were loaded with rhodamine B (RhB) dye before the SA coating to examine the pH-responsiveness of the hybrid materials. Subsequently, 100 mg of NH_2_-Gd_2_O_3_@MSN was dispersed in 40 mL of ddH_2_O with 4 mg of RhB, then the solutions was stirred for 24 h at ambient temperature. Afterward, the above protocol for the grafting of the polymers was followed again. The loading capacity (LC) of the RhB was calculated according to the following equation:(1)$$LC=\frac{m_{A}-m_{B}}{m_{A}-m_{B}+m_{C}}$$

In which *m_A_* was the added mass of the RhB, m_B_ was the mass of the RhB in the supernatant, and m_C_ was the total mass of the NH_2_-Gd_2_O_3_@MSN.

The particles (SA-Gd_2_O_3_@MSN-RhB, 5 mg) were soaked in 5 mL PBS buffer solutions (pH 7.4, 5.5, and 4.5) at ambient temperature to examine the pH-dependent releasing efficiency. The mixed solutions were transferred into a dialysis bag (n = 3 for each condition). Subsequently, the samples were maintained in buffer solution and shaken at 37 °C. At timed intervals, 3 mL of solution was withdrawn, and an equal volume of fresh buffer was added after the respective sampling. The amount of released RhB was investigated using an FL (fluorospectrophotometer). 

### 3.5. In Vitro T1 Relaxivity Evaluation

The T1 relaxivity values of SA-Gd_2_O_3_@MSN at varying concentrations (0.025, 0.05, 0.075, 0.100, 0.150, and 0.200 mM) of Gd were examined using a 0.5 T NMI20 Analyst NMR system (Niumag, Suzhou, China). The instrumental parameters were set as follows: sampling bandwidth (SW): 100 kHz; time wait (TW): 4000 ms; regulated first data (RFD): 0.08 ms; number of samples (NS): 8; echo time (TE): 2 ms.

### 3.6. Colloidal Stability

The particles were dispersed in ddH_2_O, saline, or RPMI 1640 medium with 10% FBS (fetal bovine serum) at a concentration of 2 mg of NPs/mL for the colloidal stability test. The samples were mixed via ultrasonic concussion and left for 1 min. Subsequently, the change in the particle size was recorded through DLS measurements for up to 24 h. 

### 3.7. Biocompatibility

The blood compatibility was examined through hemolysis assay experiments in accordance with previous reports [59,60]. First, fresh whole blood was taken from each volunteer, then the red blood cells (RBCs) were centrifugally isolated at 2500 rpm for 6 min at 4 ℃. Subsequently, the RBCs were purified by washing them another five times with saline. Next, 2 mL of diluted RBC (4% *v/v*) suspension was mixed with 2 mL of SA-Gd_2_O_3_@MSN NPs (50, 100, 200, 400, and 800 μg/mL) dispersed in saline. The RBCs were incubated with ddH_2_O and saline as a negative control and positive control, respectively. Next, the system was incubated at 37 ℃ for 4 h and then centrifugated at 8000 rpm for 10 min. The absorbance of the supernatants at 576 nm was examined using a UV-2450 spectrophotometer (Shimadzu, Tokyo, Japan). The hemolysis percentage was obtained as follows (repeated three times for each respective sample): (2)$$Hemolysis(\%)=\frac{\left(A_{Sample}-A_{Negative\;Control}\right)}{\left(A_{Positive\;Control}-A_{Negative\;Control}\right)}\times 100\%$$
where *A_Sample_* represents the absorption intensity of the SA-Gd_2_O_3_@MSN-treated group; *A_Negative Control_* and *A_Positive Control_* denote the absorbance of the normal saline and ddH_2_O groups, respectively.

The MCF-7 cells (HTB-22) and 4T1 cells (CRL-2539) were employed in combination with a CCK-8 assay to examine the NPs’ cytotoxicity. In brief, the cells were separately plated in a 96-well plate at a density of 6 × 10^3^ cells per well and then cultured in RPMI-1640 containing 10% FBS and 1% penicillin streptomycin at 37 ℃ under 5% CO_2_ conditions for 24 h. Next, the SA-Gd_2_O_3_@MSN NPs were added to the medium at different concentrations (0, 5, 25, 50, 100, and 200 μg/mL). After being incubated for 24 h, the original medium was removed and replaced with 100 μL of fresh medium. Subsequently, CCK-8 solution (10 μL) was added to the respective well, and the cells were further incubated for 2 h. Lastly, the absorbance at 490 nm of the respective well was examined using a plate reader (SAFIRE2, TECAN, Männedorf, Switzerland). 

Female Balb/c mice (18–20 g) were used as the animal models to examine the biotoxicity of the SA-Gd_2_O_3_@MSN NPs in vivo. All experiments gained approval from the Animal Ethics Committee of Qiqihar Medical University (No. QMU-AECC-2021-100). Saline and SA-Gd_2_O_3_@MSN NPs at a high dosage (50 mg/kg) were injected through the tail vein to the mice. After 7 days, all the mice were dissected, and the main organs (including the heart, liver, spleen, lungs, and kidneys) were collected and examined through histopathological examination (Leica-DM4B digital microscope, Hesse Germany).

### 3.8. Cellular Uptake

The cellular uptake and in vitro MR imaging capacities of the SA-Gd_2_O_3_@MSN-RhB NPs were examined using 4T1 cells. In brief, the 4T1 cells were incubated with 100 μg/mL of SA-Gd_2_O_3_@MSN-RhB NPs for different durations (6, 12, and 24 h) after being seeded into 6-well plates (1 × 10^5^ cells/well) and allowed to stand for overnight adhesion. Subsequently, the cells were washed thoroughly with PBS, fixed with 4% paraformaldehyde, and then stained with DAPI for CLSM (confocal laser scanning microscopy) observations. A quantitative evaluation of the cellular uptake was performed using FCM (flow cytometry). At specific points of time (6, 12, and 24 h), trypsinized cells (without EDTA) in the respective well were gently resuspended in 1 mL PBS and then analyzed with a FACS Canto flow cytometer (Becton, Dickinson, ND, USA).

To examine the MRI enhancement effect of the materials in cells, the 4T1 cells were inoculated on a 6-well culture plate incubated with 1 mL RPMI 1640 for 24 h. The cells were washed twice with PBS, then 1 mL of fresh media containing SA-Gd_2_O_3_@MSN-RhB NPs was introduced at a concentration of 100 μg/mL. Subsequently, the medium was removed at the preset points of time (6, 12, and 24 h). Next, the cells were digested with trypsin and then centrifuged at 1000 rpm for 6 min. Then, 0.5 mL of PBS was introduced to suspend the cells and the cells were further fixed by adding 0.5 mL of 2.0 wt% agarose solution to the centrifugation tubes. MRI scanning was performed using a Philips 3.0 T MRI scanner and a human carotid wall coil. The T1WI sequence was used as follows: TR/TE: 650 ms/10 ms; layer thickness: 3 mm; layer spacing: 2 mm; matrix: 192 × 192; FOV: 130 × 130 mm; nexus: 2 times.

### 3.9. In Vivo Proof-of-Concept Study

A 4T1 tumor model was built by injecting 100 μL of 4T1 cell suspension (1 × 10^6^ cells) into the right armpit of female Balb/c mice. To examine the behavior of the RhB-loaded SA-Gd_2_O_3_@MSN hybrid systems in acidic pH environments, the mice were randomly assigned to three groups and treated with free RhB, RhB-loaded MSNs (NH_2_-Gd_2_O_3_@MSN-RhB), and SA-Gd_2_O_3_@MSN-RhB at a dose of 200 mg/kg RhB via tail vein injection. The fluorescence images (FI) were captured at 1, 6, 12, 24, and 48 h after injection using the IVIS Lumina Ⅲ imaging system (PerkinElmer, Waltham, MA, USA). The 4T1 tumor-bearing mice were injected with 100 μL of SA-Gd_2_O_3_@MSN NPs (1 mg of Gd^3+^ per kg) via the tail vein for in vivo MR imaging. Subsequently, the MR images were captured at 0, 1, and 24 h after injection using a 3.0T clinical MRI scanner (Philips).

### 3.10. Statistical Analysis

The statistical data were investigated using SPSS 20.0 software with a two-tailed Student’s t-test. The error bars presented in this study represent the standard deviation (SD). Here, a *p*-value < 0.05 indicates a difference achieving statistical significance.

## 4. Conclusions

In this study, a imaging nanocarrier system was developed, which was functionalized with a pH-responsive polymer. The results of the colloidal stability, hemolysis assay, and toxicity experiments suggested that the SA-coated NPs can enhance the biocompatibility and stability of hybrid mesoporous silica. The in vitro and in vivo release results confirmed that SA-Gd_2_O_3_@MSN-RhB exhibits pH-responsive drug release behavior in acidic environments. Moreover, the developed multifunctional nanoparticles can serve as efficient probes for the MR imaging of tumor models in vivo. The results of this study suggest that the hybrid materials investigated here can be employed as smart drug delivery and imaging nanocarriers. However, our cell-based and animal-based explorations are not enough to represent medical applications, which require further studies to validate the efficacy of these NPs.

## Data Availability

The data presented in this study are available on request from the corresponding author.

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
