# Peer review of "pH-Responsive Drug Delivery and Imaging Study of Hybrid Mesoporous Silica Nanoparticles"

_molecules, 2022, doi:10.3390/molecules27196519_

Round 1

Reviewer 1 Report

Dear authors

I have gone in details through your manuscript and would like you to answer the following querries.

1. Please make all in vitro and in vivo words Italic.

2. Please revise the introduction section by adding information about pH-responsive carrier systems, their implications, their uses, merits, demerits, and what gap you identified that you address by using mesoporous silica nanoparticles with magnetic properties.

3. Please further elaborate your FTIR results. They are poorly interpreted. What increase in absorption peak at 1375 means? Does the wave number shifted to higher value OR absorbance? As your results did not translated into a logical justification for your claim that SA-Gd2O3@MSN were successfully prepared.

4. The reason for higher release at pH 5.5 and 4.5 are not logically explained. Reduction in pH increased the release extent, which means your particles were less retardant at acidic pH. By decreasing pH, the concentration of H ions increases. So what could be the implications with respect to H ion interaction with SA or MSN moieties? Please explain.

5. The released Rhb was estimated using UV visible technique.  Are you sure Rhb can be estimated using UV? If yes, please cite a reference. Further, was the UV method validated by estimating LOQ, LOD, percent precision and percent accuracy? Was the released Rhb amount enough which UV easily detected? Therefore, you are advised to include LOD, LOQ, percent precision and percent accuracy values for UV method you have used.

6. Please re-write the Hemolysis (%) equation using word equation function, which shall not be embedded in the text.

7. MCF-7 and 4T1 cells ATCC numbers are missing. Please add, and also add the source.

8. Does were injected into mice using which route? Please specify.

Author Response

An example can be found here.

Reviewer 2 Report

1. The developed hybrid NPS have been extensively characterized and evaluated in this study. Is the system capable of loading weakly acidic or basic drugs?

2. The loaded dose of RhB is 4mg in 100 mg of the hybrid nanoparticles. Can we load higher dose of the active components in such system?

3.      RhB, which is soluble in water, was adopted as the model drug to study the loading capacity of MSNs. Since most of the anticancer drugs show poor solubility in water, can we load such drugs in the system with the same efficacy?

4. Surface of some functionalized nanoparticles has been found to interact with proteins and other biomolecules forming protein corona that may lose the intended targeting properties of these nanoparticles. In order to rule out such interactions, can we consider such studies?

5.  Iron oxide nanoparticles (IONPs) have limited use in imaging liver, spleen and lymph nodes. Some of the FDI approved IONPs have been discontinued as MRI contrast agents in United States. Do the authors consider Mesoporous Silica Nanoparticles as safe alternative to IONPs?

6.     Did the authors investigate release of Gd+3 from the formulation without sodium alginate coating?

7. How do the authors justify flocculation of Gd2O3@MSN and NH2Gd2O3@MSN?

8.      Revision for grammatical mistakes (Line No. 133, 165, 208, 223 etc) and details of abbreviations is required.

Author Response

An example can be found here.

Reviewer 3 Report

The author presented the paper "pH-Responsive Drug Delivery and Imaging Study of Magnetic Mesoporous Silica Nanoparticles"

1) The paper has some experiments with mice. It should be don Ethics Committee document, and the number should be presented in the Institutional Review Board Statement

2) Much more 2-3 years references about such nanoparticle for therapy and diagnostics should be mentioned. The introduction does not contain a sufficient overview, and references of the state of modern research in the field of silica nanoparticles. There are a number of papers. The novelty of the work should seen from the Introduction.

3) The title does not reflect paper content. In the paper 1) no experiment with drug, only fluorescence dye; 2) no magnetic experiment, no magnetic measurement (it is MRI experiment, but not magnetic). In this way, I highly recommend to change the Title in the appropriate way. 

4) Scheme 1 doesn't show how the initial nanoparticle was obtained. What is FL?

The captions to the drawing are misleading, for example, are the different colors of the arrows in the assembly scheme fundamental? Why is there the arrows in the signature of the Scheme elements, too? 

5) Fig 1h. I can't see many changes between NH2Gd2O3MSN and Gd2O3MSN 2900 cm-1 maybe water signal, you can see it on the both pictures. Moreover, the IR spectra should be decrypted in a better way.

6) Figure  7, Why is the control after 12h, 24h have the signal? Low signal in the groups 1 and 2. No valuable discussion of this experiment. Where is the tumor in this picture?

7)  The conclusion section is poor. I highly recommend changing the Abstract and Conclusion section in the regard of the novelty, limitations, and further recommendation.

8) In the lines 136-137, the authors mentioned that "due to the weakened electrostatic interaction between the negatively charged carboxylic acid group of the SA and the positively charged MSN" However, in the references authors [36,37] authors not explained this fact. Silica nanoparticles usually have the same pH-sensitive properties in some way, which is depends on the coating and many factors. But to write that is depends only of SA I think it is wrong and have to be proved. No control release experiments were done without SA coating.

9) In the Section 2.2, the quantitative characteristics of the release of the dye are discussed, but the effectiveness of binding the dye to the carrier has not been shown anywhere before. Although capacity is one of the main characteristics of such systems.

10) Figure 4 shows the stability data. However, the stability of the systems at pH 7.4, and 5 is not shown. The dye release mechanism may depend on this data.

11) relaxivity measurenments

Gd2O3@MSN 52.48 mM-1s -1

SA-Gd2O3@MSN 11.91 mM-1s -1 Why in this composite the relativity is much lower?

Minor comments 

three word results in the abstract

83.19+-8.74 nm too precise values, the error of the experiment is high, maybe better 83.2+-8.7 and other values in the paper, too,

I recommend to divide Fig 1. into two figures. It is difficult to follow it. Moreover, Fig1e Is it DLS data number, volume, intensity (it should be mentioned).

line 116 N2

Table 1. Where are errors of the values?

Figure 4. It is necessary to move the time point 0, to the intersection of the abscissa and ordinate. Clarify, please, in the caption to the graph and / or legend that the experiments were done not in 10% FBS, but 10% FBS in RPMI 1640 medium.

Some problems lines 236, 283

ddH2O line 244 should be decrypted in the first place

Author Response

An example can be found here.

Reviewer 4 Report

This manuscript reports the preparation of Sodium Alginate-coated Gd2O3@MSN with the properties of pH responsiveness, good biocompatibility, and MR imaging, which can provide a promising drug delivery platform for MR-imaging guided diagnosis. May eventually be publishable but requires major revisions as indicated.

1.    Where is Figure 3??

2.    The diameter and morphology of NH2-Gd2O3@MSN and SA-Gd2O3@MSN are misleading, such as in Fig 1a, 1b, 1c, and Fig 4. Why do stable nanoparticles have different sizes in different solvents?

3.    It would be better to provide the uniform scale bar in the results of HE staining of major organs in Fig 5d, which can present the differences between the two groups. For example, the staining images of the heart are obviously different between the two groups.

4.    In Figure 7, the tumor site of mice should be marked. Where is the aggregation of nanoparticles in the tumor site?

5.    Although the results are organized, some more discussion should be added to improve the novelty of this work.

Author Response

An example can be found here.

Round 2

Reviewer 3 Report

The authors highly improved the paper. However, I have some minor comments

1) Scheme 1 may be better without or only with common abbreviations. For example, FI, RhB, SA are not common abbreviation. It will be easier to see the full-construction structure. Fig. S1 may be presented in the main paper, but it is for your consideration. 

2) Figure 8A. I don't clearly understand why we see the fluorescence signal for control mice after 12h. See the differences between 1h, 6h, 24h, 48 h. I think it was the problem with background correction. You can try it using your fluorescence imager program. In this way, you will have more relevant pictures without green. The same for group 1 24h, and group 24h, 48h. You can see that the places for the mouse's nose are green. It looks that green may be removed. Your pictures will be better.

3) Fig 5. It is 10% FBS in water dilution, 10%FBS in RPMI dilution, or only RPMI medium? It should be mentioned in the paper. I am a bit confused after the first version of the paper.

Author Response

An example can be found here.

Reviewer 4 Report

The authors have adequately addressed all of the comments, thus I am happy to recommend acceptance of this work for publication at Molecules.

Author Response

Thank you for your comments.

Your help will be really appreciated.

Thank you very much in advance for your careful consideration!

Best wishes,

Liguo Hao